# Effect of dexamethasone pretreatment using deep learning on the surgical effect of patients with gastrointestinal tumors

**Kun Lu, Qiang Li, Chun Pu, Xue Lei, Qiang Fu**⊙\*

Department of Anesthesiology, The Third People's Hospital of Chengdu, Southwest Jiao Tong University, Chengdu, China

\* fuqiang1878@163.com

**Data Availability Statement:** All relevant data are within the paper and its supporting information files.

## Abstract

To explore the application efficacy and significance of deep learning in anesthesia management for gastrointestinal tumors (GITs) surgery, 80 elderly patients with GITs who underwent surgical intervention at our institution between January and September 2021 were enrolled. According to the preoperative anesthesia management methodology, patients were rolled into a control (Ctrl) group (using 10 mg dexamethasone 1–2 hours before surgery) and an experimental (Exp) group (using a deep learning-based anesthesia monitoring system on the basis of the Ctrl group), with 40 cases in each group. A comprehensive comparative analysis was performed between the two cohorts, encompassing postoperative cognitive evaluations, Montreal Cognitive Assessment (MoCA) scores, gastrointestinal functionality, serum biomarkers (including interleukin (IL)-6, C-reactive protein (CRP), and cortisol levels), length of hospitalization, incidence of complications, and other pertinent metrics. The findings demonstrated that anesthesia monitoring facilitated by deep learning algorithms effectively assessed the anesthesia state of patients. Compared to the Ctrl group, patients in the Exp group showed significant differences in cognitive assessments (word recall, number connection, number coding) ($P<0.05$). Additionally, the Exp group exhibited a notably increased MoCA score (25.3±2.4), significantly shorter time to first flatus postoperatively (35.8±13.7 hours), markedly reduced postoperative pain scores, significantly shortened time to tolerate a liquid diet postoperatively (19.6±5.2 hours), accelerated recovery of serum-related indicators, and a significantly decreased mean length of hospital stay (11.4±3.2 days) compared to the Ctrl group. In summary, administering dexamethasone under the anesthesia management of GITs surgery based on gradient boosting decision tree (GBDT) and pharmacokinetics pharmacodynamics (PKPD) models can promote patient recovery, reduce the incidence of postoperative cognitive impairment (POCD), and improve patient prognosis.

## 1. Introduction

Gastrointestinal tumors (GITs) constitute a prevalent clinical entity with substantial health ramifications, spanning a diverse array of tissue types and clinical presentations. These

**Funding:** The author(s) received no specific funding for this work.

neoplasms, comprising gastric, colorectal, and esophageal cancers among others, prominently feature among digestive system malignancies on a global scale [1]. Amidst the ongoing demographic shift towards an aging population, the incidence of GITs has demonstrated a discernible upward trend in recent decades, presenting a formidable challenge within the sphere of global health [2]. Epidemiological investigations into cancer underscore a close nexus between various factors, including environmental influences, dietary modifications, genetic predisposition, and the onset of GITs [3,4]. Surgical intervention remains the cornerstone of treatment for GITs, with the primary objective of excising tumor tissue as comprehensively as feasible, thereby striving for curative or palliative outcomes. With the continuous evolution of minimally invasive surgical techniques, an expanding cohort of patients stands to benefit from procedures characterized by reduced trauma and expedited recovery [5]. Nevertheless, several factors encountered during surgery, such as patient stress responses, inflammatory processes, hypoxia, and the administration of anesthetics, have the potential to induce irreversible effects on the central nervous system, precipitating a decline in cognitive function [6]. This phenomenon assumes particular significance in elderly patients, whose cardiovascular systems inherently exhibit fragility. Heightened stress responses can trigger activation of the hypothalamic-pituitary-adrenal axis, culminating in augmented stress hormone release. Given the diminished stress tolerance in the elderly population, this may predispose individuals to postoperative cognitive dysfunction (POCD) or more severe sequelae [7]. POCD not only detrimentally impacts patients' quality of life but also extends hospitalization durations, escalates healthcare expenditures, and may even correlate with the onset of neurological disorders, thereby significantly influencing patient prognoses [8]. Consequently, the identification of efficacious preventive strategies assumes paramount importance in enhancing surgical safety, optimizing recovery trajectories, and alleviating healthcare burdens.

Dexamethasone, a synthetic long-acting glucocorticoid, possesses anti-inflammatory, anti-endotoxin, and immunosuppressive properties, rendering it a cornerstone in various clinical therapeutic regimens [9]. Notably, dexamethasone exerts a negative modulatory effect on the hypothalamic-pituitary-adrenal cortex (HPA) axis at appropriate dosages, thereby attenuating cortisol levels within the body. A plethora of clinical investigations have consistently underscored the efficacy of preoperative dexamethasone administration in significantly mitigating the incidence of postoperative nausea and vomiting, while concurrently ameliorating postoperative pain, fatigue, and other untoward patient reactions [10,11]. Nevertheless, divergent viewpoints exist within the academic domain concerning its putative impact on postoperative cognitive function. While some studies propose a potentially beneficial effect of dexamethasone in enhancing postoperative cognitive function [12], others posit a plausible association between dexamethasone administration and cognitive decline, with a conceivable elevation in the risk of POCD among patients [13]. In light of this context, conducting a comprehensive exploration of the preoperative effects of dexamethasone on postoperative cognitive function assumes paramount importance. Furthermore, meticulous monitoring of depth of anesthesia (DOA) stands as a pivotal component in safeguarding the well-being of surgical patients. Inadequate depth of anesthesia can predispose patients to intraoperative awareness, involuntary bodily movements, and other adverse reactions, thereby not only impeding the smooth conduct of surgery but potentially inflicting psychological trauma upon patients. Conversely, excessive depth of anesthesia carries the inherent risk of compromising patients' physiological function and gravely jeopardizing their lives [14]. Given the inherent limitations of manual monitoring techniques, the electroencephalogram (EEG) analysis method has emerged as a cornerstone in the realm of anesthesia monitoring [15]. Considering that anesthetic agents primarily exert their effects on the central nervous system, relying solely on clinical parameters for assessing the consciousness level of an anesthetized individual is inherently unreliable.

Consequently, the analysis of brain activity via EEG holds immense practical value in anesthesia monitoring endeavors.

In recent years, deep learning, as a pivotal branch of artificial intelligence, has achieved significant breakthroughs and advancements. Transitioning from the initial convolutional neural network (CNN) to the recurrent neural network (RNN), and subsequently to the transformative paradigm of Transformer, deep learning has continually broadened its scope of applications across domains such as image analysis, speech recognition, and natural language processing [16]. Researchers consistently introduce innovative model architectures, optimization algorithms, and pre-training strategies, addressing challenges that traditional methodologies struggled to surmount, thereby elevating computational capabilities to remarkable levels in tasks encompassing identification, generation, and reasoning. While conventional EEG analysis methods are constrained by limitations in feature extraction efficiency and pattern recognition, deep learning has facilitated more precise and automated analysis of EEG data through technologies including CNNs, RNNs, and attention mechanisms, thereby demonstrating promising prospects in realms such as brain-computer interface development, emotion recognition, and diagnosis of neurological disorders [17].

Hence, in this study, a deep learning-based anesthesia detection system was constructed and assisted in dexamethasone pretreatment for elderly gastrointestinal cancer patients. By comparing it with patients without system assistance, the impact on surgical outcomes was explored, providing scientific basis for clinical decision-making. The research aimed to provide more scientific and accurate decision-making basis for clinical practice, and improve the treatment effectiveness and quality of life of patients with GITs.

## 2. Research methods

### 2.1 Research objects

In this study, a cohort of 80 elderly patients diagnosed with GITs scheduled for surgical intervention at our institution between January and September 2021 was assembled. The patient cohort comprised 37 males and 43 females, with a mean age of (74.6 ± 3.3) years. According to the preoperative anesthesia pre-treatment methodology, participants were randomly divided into a control (Ctrl) group and an experimental (Exp) group. Patients in the Ctrl group received intravenous injection of 10 mg (2 mL) of dexamethasone 1–2 hours before surgery. Patients in the Exp group used a deep learning technology-based anesthesia detection system to monitor anesthesia status on the basis of the anesthesia preprocessing scheme of the Ctrl group. This study has obtained approval from the Medical Ethics Review Board of the Third People's Hospital of Chengdu, with approval number (2020) Ethics Review No. (Y-89). All experiments were conducted in accordance with the regulations of relevant departments.

### 2.2 Criteria for patients to be included or excluded

The inclusion criteria for patient selection were defined as follows: i) age $\geq$ 65 years old; ii) American Society of Anesthesiologists (ASA) classification II—III; iii) confirmed diagnosis of GITs via postoperative pathological examination; and iv) patients who were fully informed of the study's objectives, voluntarily opted to participate, and provided signed informed consent.

Patients were excluded from the study if they met any of the following criteria: i) presence of mental illness or inability to cooperate with study procedures; ii) diagnosis of intestinal obstruction or active gastric ulcer; iii) known allergy to dexamethasone or any systemic steroid; iv) recent use of steroids within the past 3 months; and v) history of abdominal surgery.

## 2.3 Preoperative anesthesia methods

All patients scheduled for elective surgery adhered strictly to a preoperative fasting regimen, refraining from food intake for twelve hours and abstaining from liquids for six hours prior to the surgical procedure. Additionally, all patients received an intramuscular injection of 0.5 mg atropine 30 minutes before induction of anesthesia. Upon arrival in the operating room, peripheral intravenous access was established, and a solution of sodium lactate Ringer's fluid was administered at a regulated infusion rate. Subsequently, a multifunctional vital signs monitor was employed to continuously monitor parameters including blood pressure (BP), heart rate (HR), electrocardiogram (ECG), pulse oxygen saturation (SPO$_2$), and partial pressure of end-tidal carbon dioxide (PETCO$_2$). Following this, the Bispectral index monitor (BIS) was subsequently connected for additional monitoring.

Prior to anesthesia induction, patients underwent inhalation of pure oxygen via mask for a duration of 5–6 minutes to ensure optimal oxygenation and nitrogen elimination. Anesthesia induction was initiated through intravenous administration of midazolam at a dosage range of 0.02–0.03 mg/kg, fentanyl at 2–3 μg/kg, propofol at 1.0–1.5 mg/kg, and rocuronium at 0.6–0.8 mg/kg. Upon cessation of spontaneous respiration, controlled mask ventilation was initiated using a ventilator, followed by endotracheal intubation after 3 minutes. Subsequently, mechanical ventilation commenced upon successful connection to the ventilator. The ventilator was operated in volume-controlled ventilation mode, with respiratory parameters set as follows: respiratory rate at 12 breaths per minute, tidal volume ranging from 6–8 mL/kg, and an inspiration-to-expiration ratio of 1:2. Following this, immediate catheterization was performed via puncture of the right internal jugular vein and radial artery.

General anesthesia was maintained via a combination of intravenous and inhalation agents, with continuous intravenous infusion of propofol and remifentanil, alongside inhalation of sevoflurane. Intraoperative monitoring of invasive arterial blood pressure was conducted continuously to ensure fluctuations did not exceed 25% of the baseline value. Additionally, PaCO$_2$ was maintained within the range of 35–45 mmHg, while BIS was regulated between 40–60 through adjustment of inhalational anesthetic concentration. Throughout the surgical procedure, fluid replenishment rates were adjusted based on comprehensive assessment of hemodynamic parameters such as blood pressure and heart rate. In instances where heart rate remained below 50 beats per minute despite adjustments in anesthesia depth and fluid rehydration, intravenous administration of 0.25 mg atropine was administered. Moreover, if systolic blood pressure (SBP) fell below 90 mmHg, intravenous injection of 6 mg ephedrine was administered, whereas if SBP exceeded 160 mmHg, 12.5 mg urapidil was intravenously administered.

After the end of anesthesia, the anesthetic drug was stopped. After the patient recovered completely, the tracheal catheter was removed and safely returned to the ward.

## 2.4 BIS detection based on deep learning

Based on deep learning algorithm, the regression prediction of BIS index is realized through EEG data.

(I) Gradient boosted decision tree (GBDT)

Boosting tree is a lifting method based on decision tree. In classification problems, decision trees are commonly employed as the basis function, facilitating the classification of data points into distinct categories or classes. Conversely, in regression problems, the basis function typically utilizes a regression tree, enabling the prediction of continuous numerical values based on input features. First, the basic function is initialized to make $f_0(x) = 0$, and the model of step

$m$ is shown in Eq (1):

$$f_m(x) = f_{m-1}(x) + T(x; \Xi_m) \tag{1}$$

In Eq (1), $f_{m-1}(x)$ is the current model, and the parameter $\xi_m$ is determined by solving the minimum value of the loss function, as shown in Eq (2):

$$\Xi_m = \arg \min_{\Xi_m} \sum_{i=1}^{N} L(y_i, f_{m-1}(x_i) + T(x; \Xi_m)) \tag{2}$$

For the lifting tree algorithm of regression problem, the forward step-by-step algorithm is used, as shown in Eqs (3–5):

$$f_0(x) = 0 \tag{3}$$

$$f_m(x) = f_{m-1}(x) + T(x; \Xi_m), m = 1, 2, \ldots, M \tag{4}$$

$$f_M(x) = \sum_{m=1}^{M} T(x; \Xi_m) \tag{5}$$

In the $m$-th step of the forward step-by-step algorithm, the parameter $m$ in Eq (2) is obtained by the demand solution. When the square error is used as the loss function, it is calculated as shown in Eq (6):

$$L(y, f(x)) = (y - f(x))^2 \tag{6}$$

$$L(y, f_{m-1}(x) - T(x; \Xi_m)) = (y - f_{m-1}(x) - T(x; \Xi_m))^2 = [r - T(x; \Xi_m)]^2 \tag{7}$$

In Eq (7), $r$ is the residual obtained by fitting the data under the current model.

$$r = y - f_{m-1}(x) \tag{8}$$

When employing the GBDT algorithm to address regression problems, if the loss function is defined as square loss, the optimization process remains relatively straightforward, requiring solely the fitting of the model residuals. However, when the loss function adopts a more generalized form, the optimization procedure becomes significantly more intricate. To address this challenge, Friedman proposed leveraging the negative gradient value of the loss function as an approximation of the residual, and subsequently fitting the regression tree via gradient boosting, thus giving rise to the GBDT algorithm. At the heart of the GBDT algorithm lies the amalgamation of m weak classifiers into a robust classifier utilizing Boosting principles, with each regression tree learning the residuals of all preceding regression trees. Through iterative refinement, the model's residual is continually diminished, optimizing the model outcomes. The gradient of residual reduction is determined by solving for the minimum value of the loss function, and establishing a new model in the direction of the gradient, as illustrated in Eq (9):

$$L = \frac{1}{2[y_i - f_i(x)]^2} \tag{9}$$

In Eq (9), $L$ is the loss function, $y_i$ is the true value, and $f_i(x)$ is the estimated value of the model. To use the GBDT model to forecast, it first inputs a sample instance and gives it a constant value. Then, the value of the negative gradient of the loss function is used as the residual estimation of the current model. The residual is fitted by regression tree, and the loss function is minimized. Finally, the regression tree is iteratively updated to get the final model $f(x)$.

(II) Pharmacokinetics model

A model capable of accurately and effectively delineating the drug metabolism dynamics during anesthesia constitutes a pivotal foundation for achieving closed-loop control of anesthesia. In 1979, Sheiner introduced the PKPD (pharmacokinetic-pharmacodynamic) model, which elucidates drug metabolism processes. The PK (pharmacokinetic) model encompasses three compartments: the central chamber, rapid distribution chamber, and slow distribution chamber, primarily elucidating the correlation between drug injection rate and drug concentration. This relationship is often expressed through third-order differential equations, as depicted in Eq (10):

$$\begin{cases} \dfrac{dC_1(t)}{dt} = -(k_{10} + k_{12} + k_{13})C_1(t) + k_{21}\dfrac{V_2 C_2(t)}{V_1} + k_{31}\dfrac{V_3 C_3(t)}{V_1} + \dfrac{u(t)}{V_1} \\[2mm] \qquad\qquad \dfrac{dC_2(t)}{dt} = k_{12}\dfrac{V_1 C_1(t)}{V_2} - k_{21}C_2(t) \\[2mm] \qquad\qquad \dfrac{dC_3(t)}{dt} = k_{13}\dfrac{V_1 C_1(t)}{V_3} - k_{31}C_3(t) \end{cases} \tag{10}$$

In Eq (10), $C_1$, $V_{ij}(i = 1,2,3)$, and $C_3$ represent the drug concentration in three atrioventricular compartments, respectively, and $C_2(t)$ represents the drug infusion rate; $K_{ij}(i{\neq}j)$ is the transmission rate of drugs from atrioventricular $i$ to atrioventricular $j$, and $K_{ij}$ is determined by parameters such as the patient's height, weight, sex, and age, as shown in Eqs (11–15):

$$k_{10} = \frac{C_{l1}}{V_1}, k_{12} = \frac{C_{l2}}{V_1}, k_{13} = \frac{C_{l3}}{V_1}, k_{21} = \frac{C_{l2}}{V_2}, k_{31} = \frac{C_{l3}}{V_3} \tag{11}$$

$$V_1 = 4.27, V_2 = 18.9 - 0.391 \times (age - 53), V_3 = 238 \tag{12}$$

$$C_{l1} = 1.89 + 0.0456 \times (weight - 77) - 0.0681 \times (lbm - 59) + 0.0264 \times (height - 177) \tag{13}$$

$$C_{l2} = 1.29 - 0.024 \times (age - 53) \tag{14}$$

$$C_{l3} = 0.836 \tag{15}$$

Where, $V_{ij}(i = 1,2,3)$ is the volume of each atria; $C_{l1}$, $C_{l2}$, and $C_{l3}$ are the elimination rates of drugs (dexamethasone in this study) in three atria, and they are related to the height, weight, sex, and age of patients. The calculation equation of *lbm* is shown in Eq (16):

$$lbm = \begin{cases} 1.07 \cdot weight - 148 \cdot \dfrac{weight^2}{height^2}, female \\[3mm] 1.1 \cdot weight - 128 \cdot \dfrac{weight^2}{weight^2}, male \end{cases} \tag{16}$$

The PD model describes the relationship between the drug effect and the concentration in the effect room, as shown in Eq (17):

$$BIS(t) = E_0 - E_{max1} \cdot \frac{C_e(t)^{\gamma 1}}{(EC_{50\_B})^{\gamma 1} + C_e(t)^{\gamma 1}} \tag{17}$$

In Eq (17), $E_0$ and $E_{max1}$ are the minimum and maximum values of BIS, respectively, and $EC_{50\_B}$ is half of the maximum value of BIS, and other parameters in the equation are influenced by the physiological parameters of the patient.

Due to the inherent inter-individual variabilities among patients, a uniform set of parameter standards cannot be universally applied when adopting the PD model. Hence, the PSO algorithm has been employed for parameter identification of the PD model. The PSO algorithm belongs to the category of evolutionary algorithms, renowned for its rapid convergence, high precision, and straightforward implementation. Throughout the solving process, the algorithm commences from a random solution and iteratively navigates towards the optimal solution through cyclic iterations. Moreover, the algorithm demonstrates exceptional performance in practical applications. The specific procedural workflow is as follows:

In the $K$-dimensional search space, there are $m$ particles, and the $i$-th particle can be expressed as:

$$X_i = (x_{i1}, x_{i2}, \ldots, x_{iK}), i = 1, 2, \ldots, m \tag{18}$$

The velocity $V_i$ and position $L_i$ of the $i$-th particle are expressed as shown in Eq (19):

$$V_i = (v_{i1}, v_{i2}, \ldots, v_{iK}), i = 1, 2, \ldots, m$$

$$L_i = (l_{i1}, l_{i2}, \ldots, l_{iK}), i = 1, 2, \ldots, m \tag{19}$$

The global optimal position $L_g$ currently searched by the group is expressed as shown in Eq (20):

$$L_g = (l_{g1}, l_{g2}, \ldots, l_{gK})^T \tag{20}$$

Particle update speed and position according to Eqs (21) and (22), respectively:

$$v_{ik} = \omega \cdot v_{ik} + C_1 \cdot rand_1 \cdot (l_{ik} - x_{ik}) + C_2 \cdot rand_2 \cdot (l_{gk} - x_{ik}), i = 1, 2, \ldots, m \tag{21}$$

$$x_{ik} = x_{ik} + v_{ik}, i = 1, 2, \ldots, m \tag{22}$$

In Eq (21), $C_1$ and $C_2$ are learning factors. *rand1* and *rand2* are random numbers between (0,1). When the number of iterations of the algorithm reaches the upper limit of the set number of iterations, the iteration is stopped and the optimal solution in the current state is output.

(III) Evaluation of results

In this study, $R^2$ coefficient, RMSE, and MAPE are used to evaluate the detection results, and the calculation equations are as follows.

$$R^2 = 1 - \frac{SSE}{SST} \tag{23}$$

$$\text{RMSE} = \sqrt{\frac{\sum_{i=1}^n r_i - \bar{r}}{n}} \tag{24}$$

$$\text{MAPE} = \sum_{i=1}^n \left| \frac{observet_t - predicted_t}{observet_t} \right| \times \frac{100}{n} \tag{25}$$

Where, $n$ is the number of samples, *SST* (Sum of Squares for total) is the sum of squares of deviations, and *SSE* (Sum of Squares for Error) is the sum of squares of residuals. *MAPE* (mean absolute percentage error) is the average percentage error. $R^2$ coefficient represents the goodness of fit of the model, and the range is (0–1). The closer it is to 1, the closer it is to the real value predicted by the model, that is, the more accurate the PD parameters identified.

*RMSE* represents the mean square error of the model, and the lower the value, the lower the error of the model. *ri* is the test sample value, and $\bar{r}$ is the average value of *r*.

In this study, a cohort of ten patients was randomly sampled, and data samples were collected accordingly. The test data were evaluated by PSO algorithm, and the PD model with BIS as the index of anesthesia depth was obtained. The parameters in Eq (17) can be obtained by PSO algorithm. According to Eqs (23) and (24), the $R^2$ coefficient and RMSE of the identification result can be calculated by comparing and analyzing the real BIS value obtained from offline files with the BIS calculated after PD model identification, and then the accuracy of BIS index identification of PD model can be obtained.

## 2.5 Postoperative observation indexes

Cognitive function assessment was conducted by a trained anesthesiologist both preoperatively and on postoperative day 7. Four specific tests were administered: a word memory test to evaluate memory; a number connection test to assess concentration and orientation; a number coding test to measure attention and visual spatial abilities; and a number repetition test to gauge attention span. Two versions of each test were employed, denoted as A and B, with version A administered preoperatively and version B postoperatively. I. The word memory test proceeded as follows: participants were instructed to sequentially read aloud 12 Mandarin words, with each word presented for 2 seconds. The reading pace was required to be steady and pronunciation clear. Following the presentation of all words, participants were prompted to recall and repeat the words heard. One point was awarded for each correct response. This process was repeated three times consecutively. II. The number connection test consisted of two variations, A and B. In version A, participants were tasked with connecting a series of numbers (1–25) on a diagram in sequential order, with the time taken to complete all connections recorded. In version B of the number connection test, participants were instructed to sequentially connect numbers on a diagram in order while also converting circles to triangles. The time taken to complete all lines was recorded. III. The number coding test involved participants reading a table and independently completing the blanks. If participants were unable to comprehend the test, a score of 0 was assigned. IV. The number repetition test encompassed two components: forward and reverse tests. In the forward test, participants were required to read a series of numbers and repeat them aloud within one second of reading. If errors were made in the first column, participants proceeded to the second column. If errors occurred in both columns, the test was terminated. Scores were allocated based on the point value assigned to each number, with the total score calculated by summing all scores. In the reverse test, participants were prompted to repeat the numbers in reverse order, starting from the end and working towards the beginning, with procedures otherwise identical to the forward test.

The Montreal cognitive assessment (MoCA) [18] was employed to evaluate POCD in patients. MoCA assesses various cognitive domains encompassing executive function, memory, attention, language, visuospatial abilities, calculation, and orientation. Scores range from 30 points, with higher scores indicating better cognitive function, while scores below 24 suggest potential cognitive impairment.

The recovery of gastrointestinal function was quantified based on the duration until the restoration of gastrointestinal motility. Postoperative pain intensity was evaluated at 24, 48, and 72 hours post-surgery using the Numeric Rating Scales (NRS) scores [19]. The NRS is a widely adopted assessment tool that measures pain levels on a scale ranging from 0 to 10, where 0 represents no pain and 10 signifies the most intense pain imaginable. The scoring criteria are as follows: "0" denotes absence of pain, "1–3" corresponds to mild pain, "4–6" indicates moderate

pain, and "7–10" reflects severe pain. For statistical analysis, scores were categorized as follows: "0" for no pain, "1" for mild pain, "2" for moderate pain, and "3" for severe pain.

Peripheral venous blood was collected and tested before surgery and at 8:00 a.m. on the third day, respectively. The serum was centrifuged at 4,000 g for 15 min and stored in a -80˚C refrigerator. Serum levels of interleukin (IL) -6, C-reactive protein (CRP), and cortisol were measured by time analysis of first defecation after chemiluminescence immunization.

Other indicators included length of stay (LOS), infection rate, incidence of digestive tract fistula, and health-related quality of life (as assessed by EQ-5D-3L) [20].

## 2.6 Statistical methods

Excel 2016 was utilized to record and summarize data. SPSS 20.0 was employed for data statistics and analysis. Mean ± standard deviation (X ± S) represented measurement data, and t test was used. Percentage (%) was the representation of count data, and X2 test was adopted. $P<0.05$ was considered to have statistical difference.

## 3. Research results

### 3.1 PD model evaluation results

The experimental results are shown in Table 1. The highest goodness of fit was 94.14(%), the lowest was 74.14(%), the highest RMSE was 8.93, and the lowest was 6.05. There are 3 patients with goodness of fit over 90(%), 7 patients with goodness of fit over 80(%), and only 3 patients with goodness of fit below 80(%). From Fig 1, the fitting effect of the model was not very good at the initial stage of the data and when the fluctuation amplitude was large, but from the point of view of statistics and clinical application, the PD model identified by PSO algorithm can fit the BIS curve well as a whole, and the anesthesia state can be evaluated under normal circumstances.

### 3.2 Comparison of cognitive function

The statistical results disclosed no great differences in the scores of word memory, number connection, number repetition, and number coding between patients from the Ctrl and Exp groups before surgery ($P > 0.05$). After the surgery, the word memory score was (13.5 ± 4.2), the number connection score was (250.3 ± 53.8), and the number coding score was (20.2 ± 5.3) in the Ctrl group; while those in the Exp group were (17.8 ± 3.9), (214.2 ± 49.4), (23.7 ± 6.1), respectively. The differences in the above three indexes presented $P<0.05$, as demonstrated in Fig 2. In Fig 3, POCD occurred in only 5 patients (12.5%) in Exp group and 12 patients (30%) in Ctrl group after surgery, showing statistically great difference ($P<0.05$).

**Table 1. PD model evaluation results.**

| Patient No. | $R^2$(%) | RMSE |
|---|---|---|
| 1 | 88.37 | 6.05 |
| 2 | 75.29 | 7.14 |
| 3 | 88.94 | 6.87 |
| 4 | 83.62 | 7.02 |
| 5 | 90.08 | 8.62 |
| 6 | 90.04 | 8.18 |
| 7 | 94.14 | 8.93 |
| 8 | 75.91 | 7.26 |
| 9 | 76.28 | 8.42 |
| 10 | 85.26 | 8.58 |

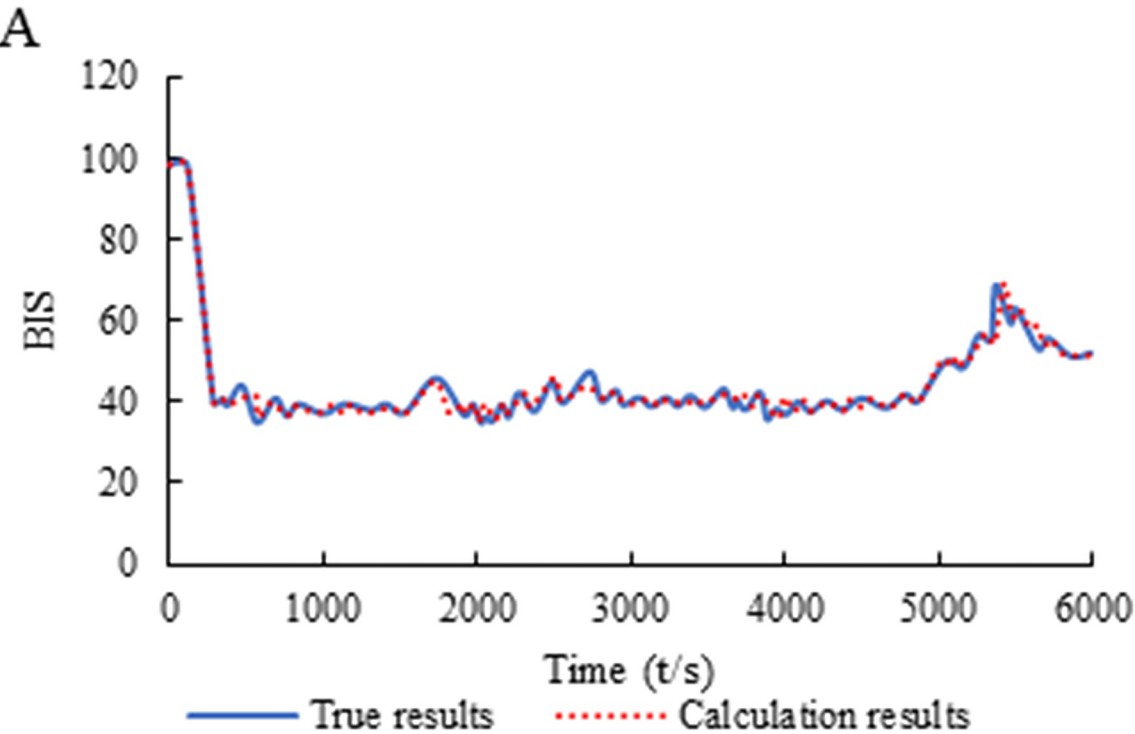

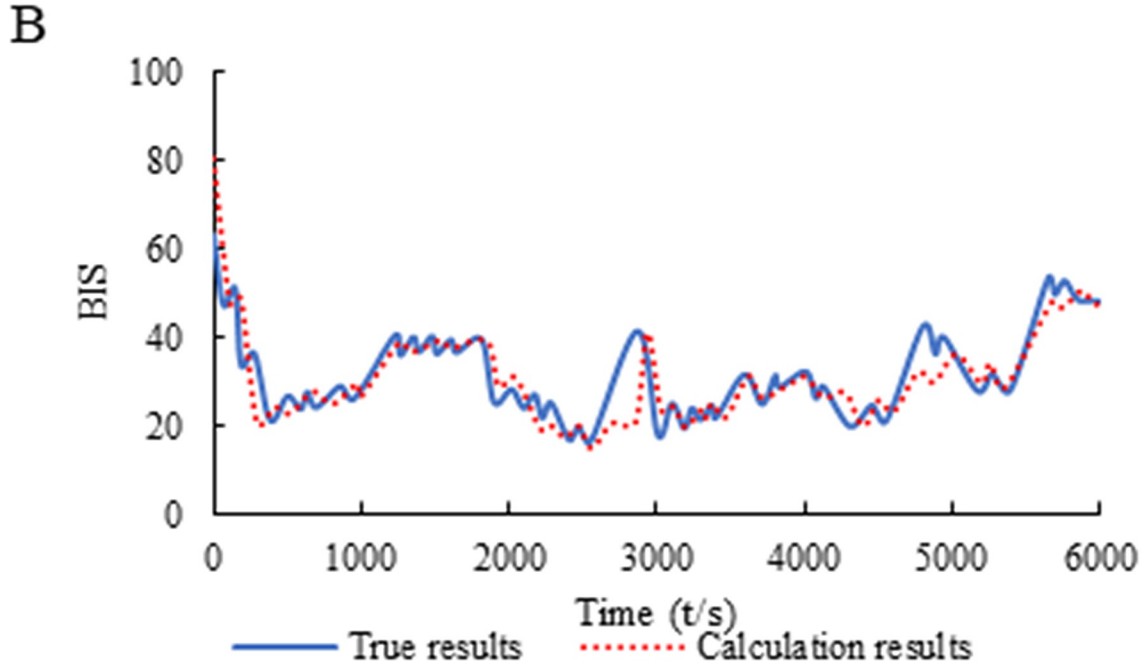

**Fig 1.** Data sample (A: Optimal $R^2$; B: Worst $R^2$).

### 3.3 MoCA rating results

The MoCA score results showed that the Ctrl group (22.8 ± 3.7) had significantly lower scores than the Exp group (25.3 ± 2.4) ($P < 0.05$) (Fig 3).

### 3.4 Comparison of gastrointestinal function recovery

According to statistics, the postoperative FET of patients in the Exp group was (35.8 ± 13.7) h, that in the Ctrl group was (46.3 ± 15.1) h. Thus, the postoperative FET in the Exp group was

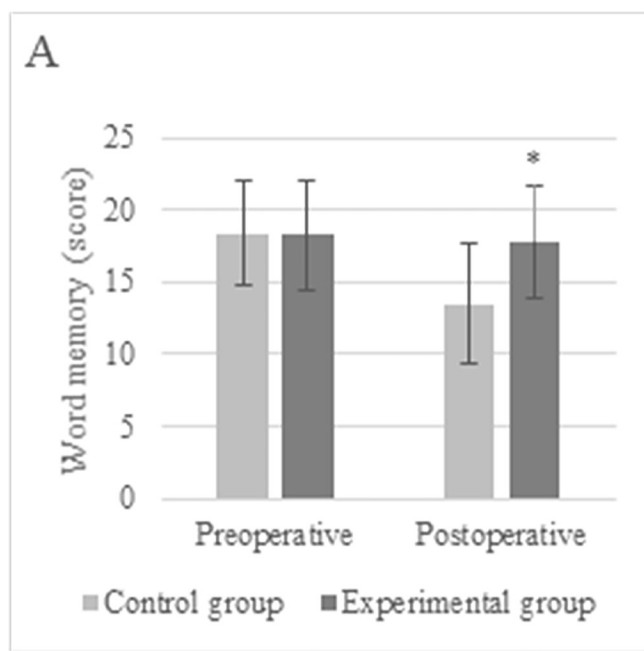

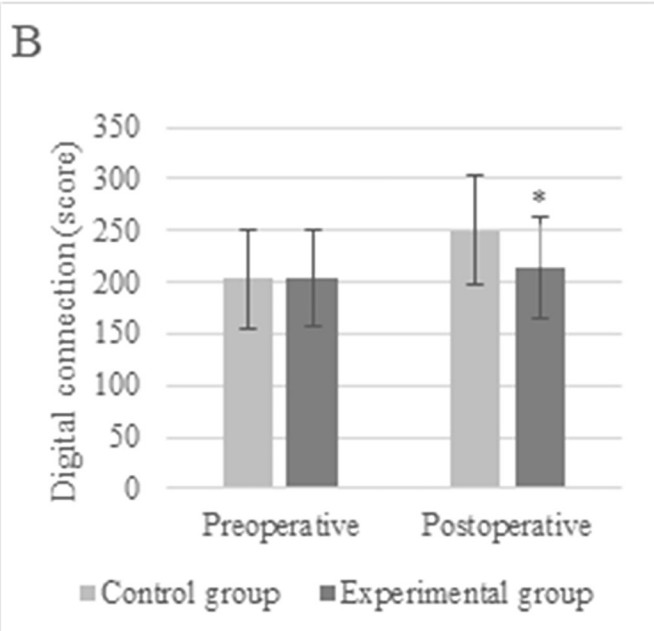

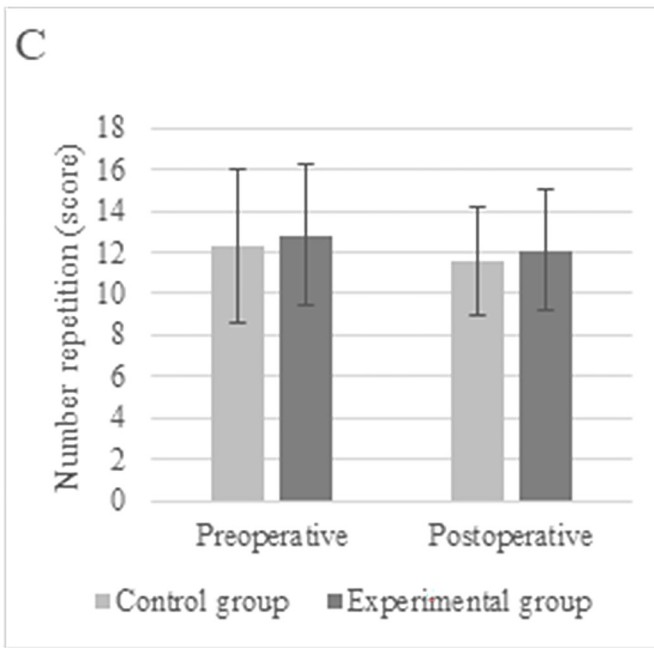

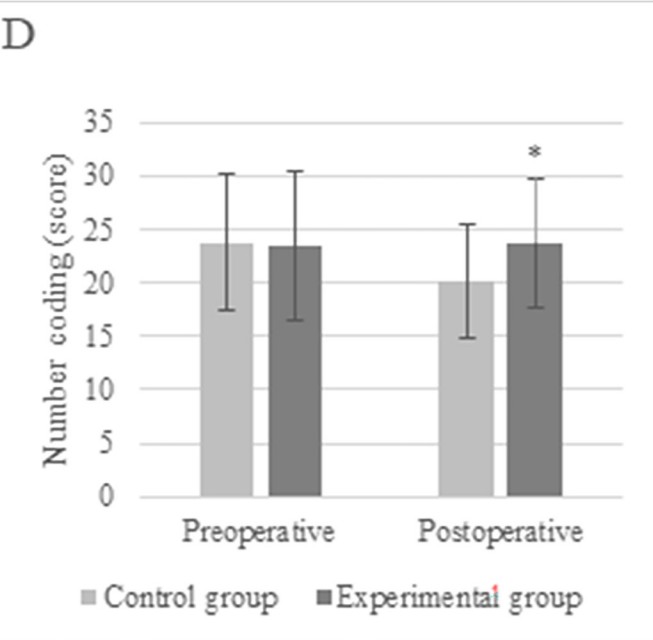

**Fig 2. Comparison of cognitive item scores of patients with different anesthesia methods (\* indicated difference with *P*<0.05).**

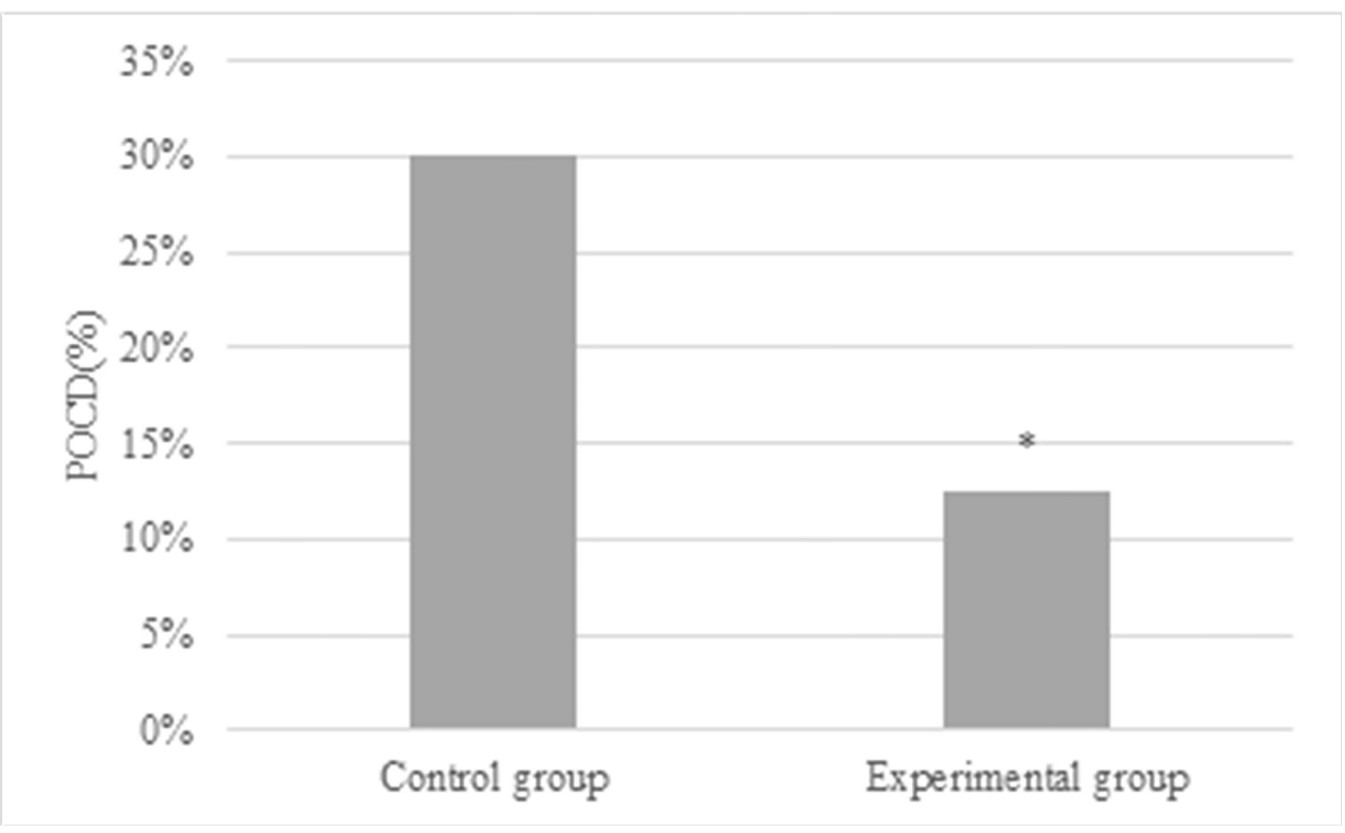

**Fig 3. Comparison of MoCA scores between two groups of patients (* indicated difference with *P*<0.05).**

extremely shortened (*P*<0.05), and the details were given in Fig 4. Fig 5 compared the pain scores of patients in different groups. The pain score in the Ctrl group was (11.3 ± 2.5), (7.4 ± 1.7), and (2.6 ± 0.5) points 12, 24, and 72 hours after surgery; while those were (6.2 ± 1.8) points, (2.8 ± 0.9) points, and (0.3 ± 0.1) points, respectively in the Ctrl group. The pain scores in the Ctrl group were much higher than those in the Exp group, exhibiting differences with *P*<0.05. The TTLD values in the Exp group and Ctrl group were (19.6 ± 5.2) hours and (23.5 ± 6.4) hours, respectively. Notably, in the Exp group, postoperative recovery time was significantly reduced compared to the Ctrl group (*P*<0.05) (Fig 6).

### 3.5 Serum IL-6, CRP, and cortisol concentrations

Serum test results suggested no observable differences in serum IL-6, CRP, and cortisol concentrations between patients in the Ctrl and Exp groups before the surgery (*P* > 0.05). On the third day after surgery, the IL-6, CRP, and cortisol concentrations in serum of patients in the Ctrl group patients were (34.6 ± 10.4) pg/mL, (7.4 ± 2.2) mg/dl, and (724.8 ± 336.3) pg/mL, respectively; while those in the Exp group were (27.5 ± 8.8) pg/mL, (5.3 ± 3.9) mg/dl, and (51.3 ± 329.5) pg/mL, respectively. The recovery rate of serum related indexes in Exp group patients was effectively accelerated, showing difference with *P*<0.05 (Fig 7).

### 3.6 Comparison of other indexes

The mean LOS was (11.4 ± 3.2) days in the Exp group and (13.4 ± 3.8) days in the Ctrl group. Therefore, the mean LOS was greatly shortened in Exp group (*P*<0.05) (Fig 8). The Exp group

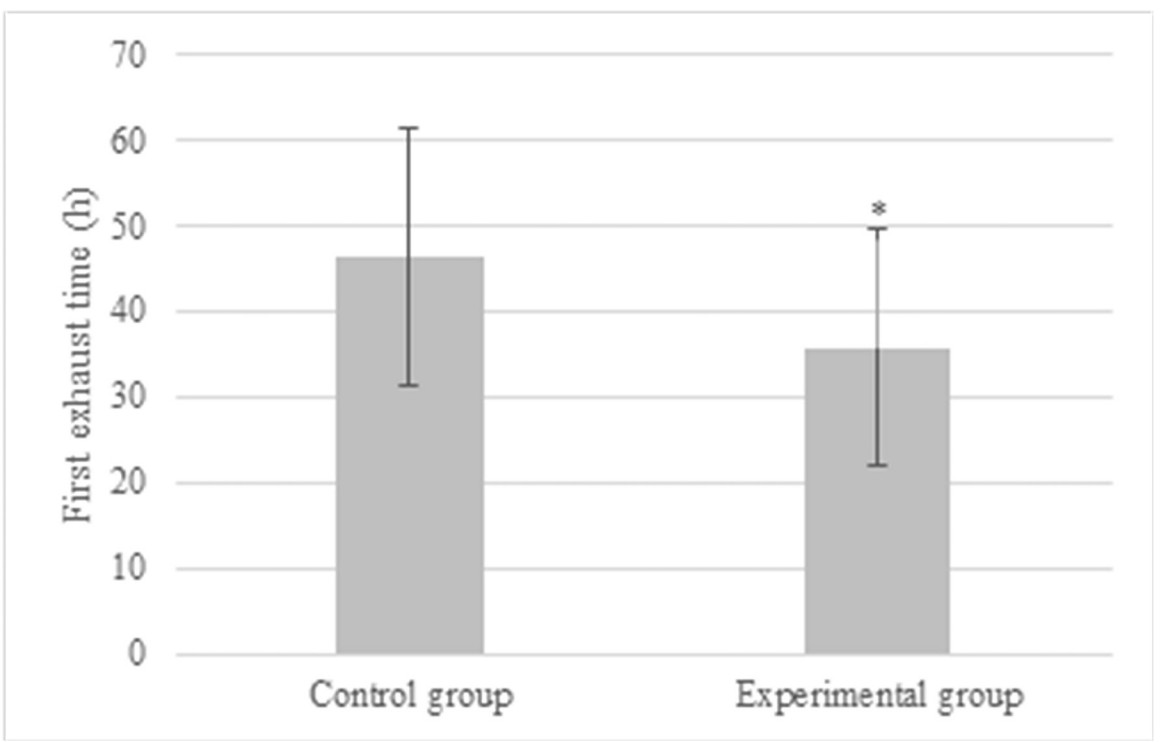

**Fig 4. Comparison of postoperative FET of patients with different anesthesia methods (* indicated difference with $P<0.05$).**

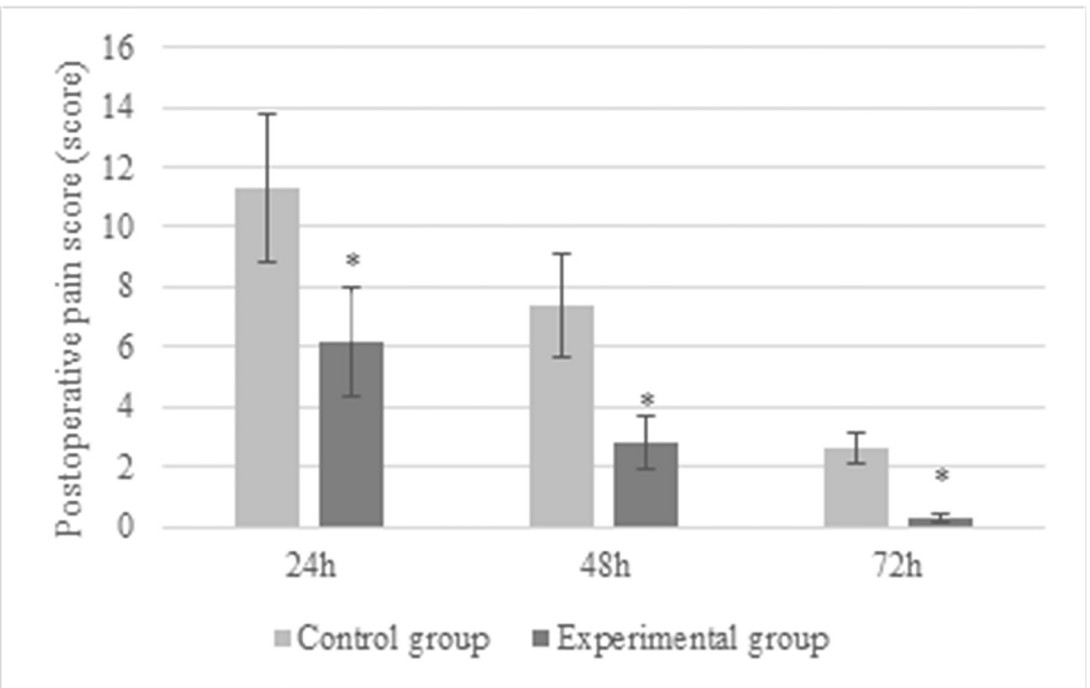

**Fig 5. Comparison of postoperative pain scores of patients with different anesthesia methods (* indicated difference with $P<0.05$).**

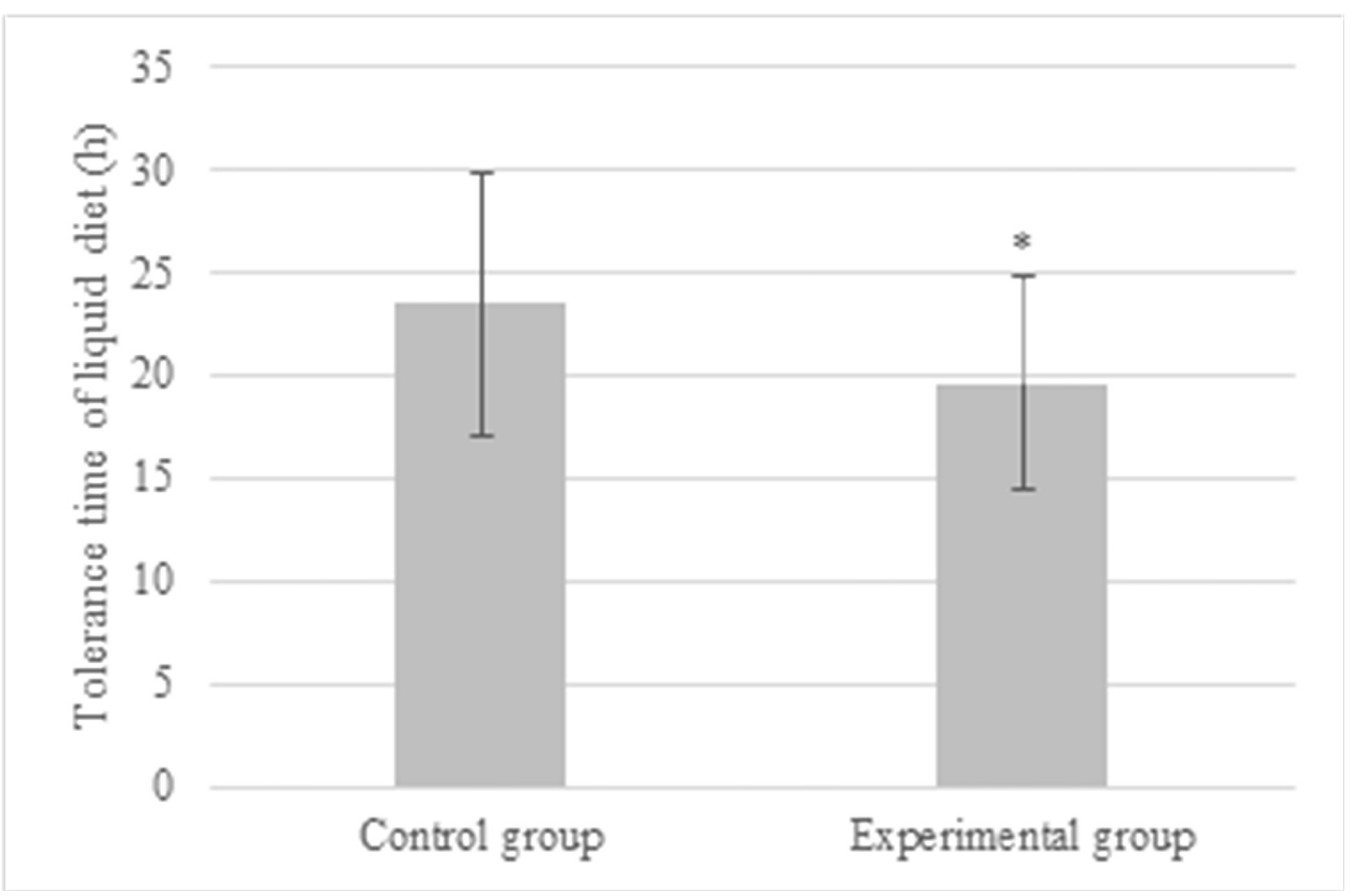

**Fig 6. Comparison of TTLD values of patients from different groups (\* indicated difference with *P*<0.05).**

patients had 4 infections and 2 gastrointestinal fistulae, while the Ctrl group patients had 5 infections and 2 gastrointestinal fistula ($P > 0.05$), as compared in Fig 9. Fig 10 suggested that the QoL scores in the Exp and Ctrl group were (93.4 ± 6.2) points and (93.1 ± 5.8) points, respectively, showing difference with $P > 0.05$.

## 4. Discussion

As an integral component of the comprehensive treatment regimen for GITs, surgical intervention plays a pivotal role in not only excising the primary lesion and reducing tumor burden but also enhancing patient prognosis to a certain extent. Nonetheless, the surgical process often elicits pain, anxiety, and physiological reactions in patients, which may detrimentally affect both surgical success and patient recovery. Thus, ensuring seamless surgical procedures and patient safety has underscored the burgeoning importance of preoperative anesthesia pretreatment [21]. This strategic approach is aimed at alleviating patient discomfort and anxiety during surgery while mitigating associated risks and complications. By implementing appropriate anesthesia techniques and pharmacological management, preoperative anesthesia pretreatment not only diminishes patients' perception of pain but also alleviates physiological stress and psychological burdens during surgery, thereby augmenting surgical success rates and enhancing postoperative recovery quality. In recent years, the advent of deep learning technology has introduced novel avenues and methodologies for optimizing anesthesia processes and ensuring safety. This technology has the capability to analyze patients' physiological

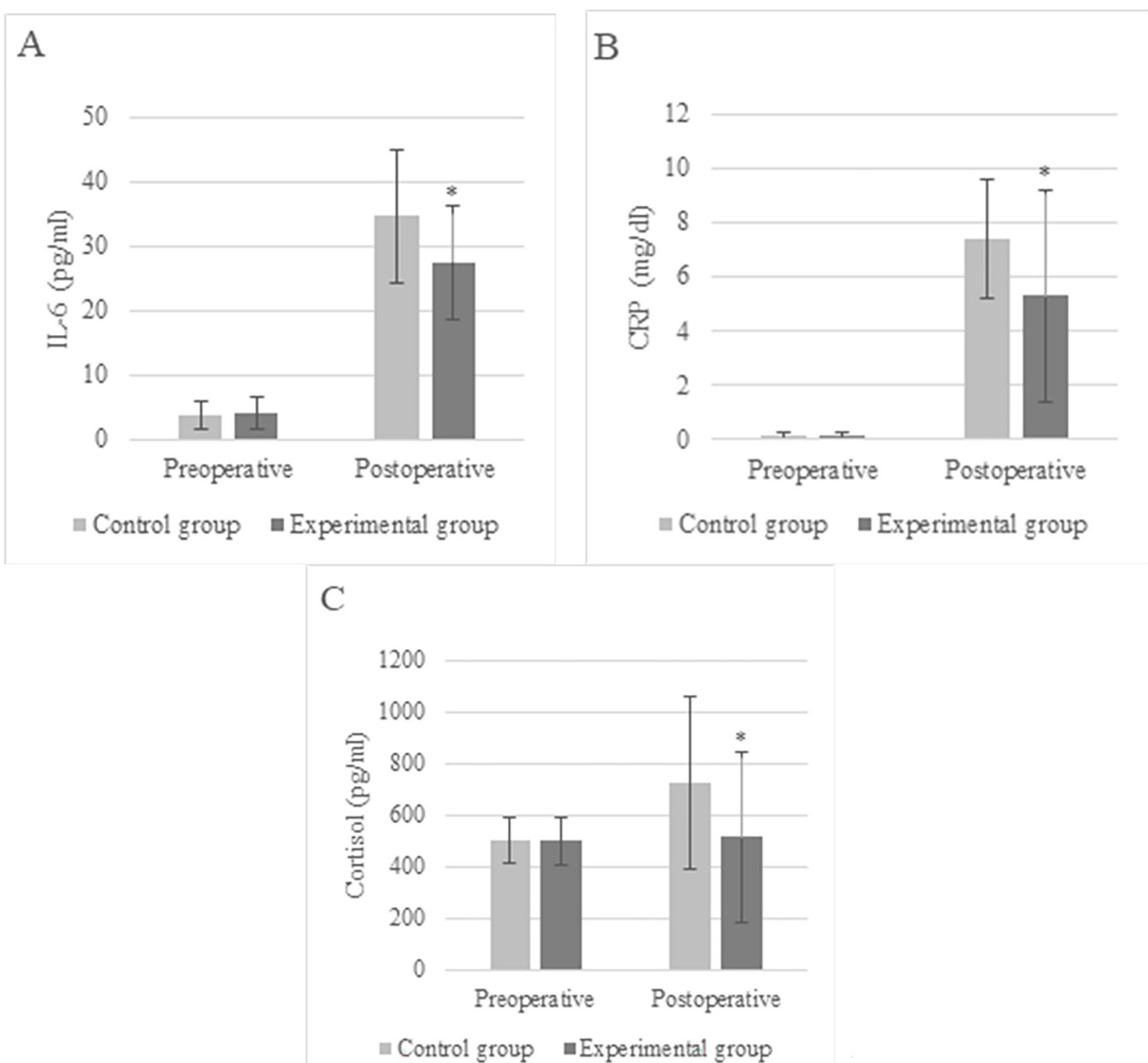

**Fig 7. Comparison of serum indexes (* indicated difference with $P<0.05$).**

signals, such as electrocardiogram and respiratory data, enabling intelligent intraoperative monitoring and drug dosage adjustments while mitigating human error and providing early warnings for potential complications [22]. In this study, leveraging a deep learning algorithm facilitated the regression prediction of BIS using EEG data. The findings demonstrated that the PD model based on the deep learning algorithm exhibited robust fitting to the BIS curve overall, effectively enabling the evaluation of anesthesia depth.

POCD manifests as a subtle decline in cognitive function following anesthesia and surgery. Presently, diagnosis relies solely on neuropsychological tests, necessitating comparisons between preoperative and postoperative states. Early-onset POCD typically manifests 1 to 2

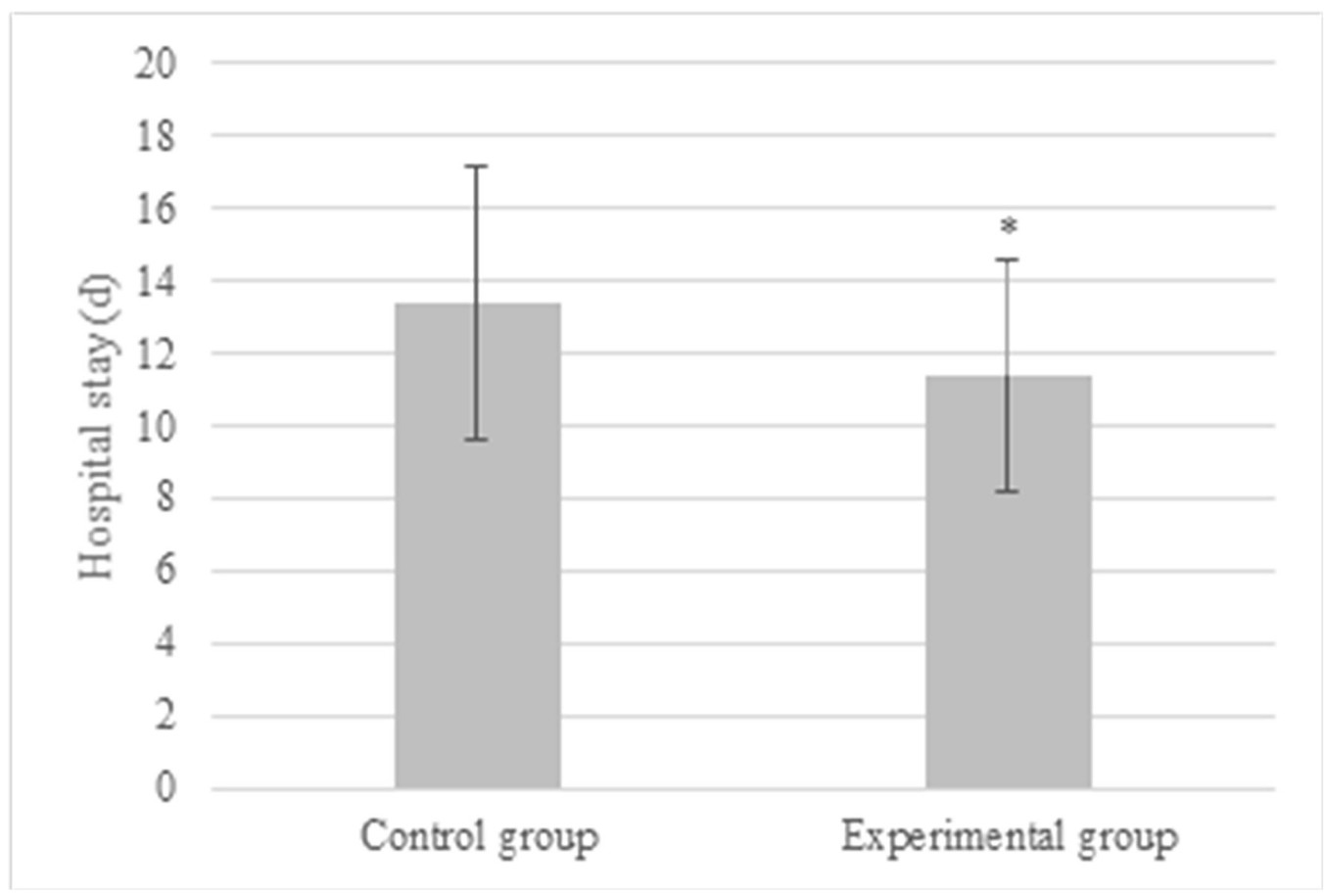

**Fig 8. LOS of patients receiving different preoperative pretreatments (* indicated difference with $P<0.05$).**

weeks post-surgery, with most cases exhibiting transient and reversible symptoms. However, some patients experience prolonged courses lasting months, years, or even permanent cognitive impairment [23]. In this study, four cognitive function tests were employed to assess memory, attention, orientation, and visual spatial abilities in patients. The exact pathogenesis of POCD remains unclear. However, existing literature suggests that its pathological onset is linked to systemic and central inflammatory responses triggered by surgery [24]. Compared with the Ctrl group, the MoCA score of the Exp group patients significantly increased ($P<0.05$), indicating that the POCD situation of the Exp group patients was significantly better than that of the Ctrl group. A foreign study revealed that although the current study did not reach statistical significance, prophylactic administration of dexamethasone seemed to help prevent the development of POCD after cardiac surgery [25], which keeps in line with the results of this work. However, a domestic meta-analysis showed that prophylactic dexamethasone did not reduce the incidence of POCD [26]. Therefore, to make progress in this area, there is still a need to test alternative prevention strategies for POCD and POD and to better understand the pathophysiology of these complex syndromes.

Early recovery of gastrointestinal function after gastrointestinal surgery is clinically important because it reduces the incidence of postoperative ileus (POI). POI is associated with postoperative pain, increased nausea and vomiting, and increased treatment costs [27]. This work confirmed that FET in Exp group patients was much shortened compared with the FET in the

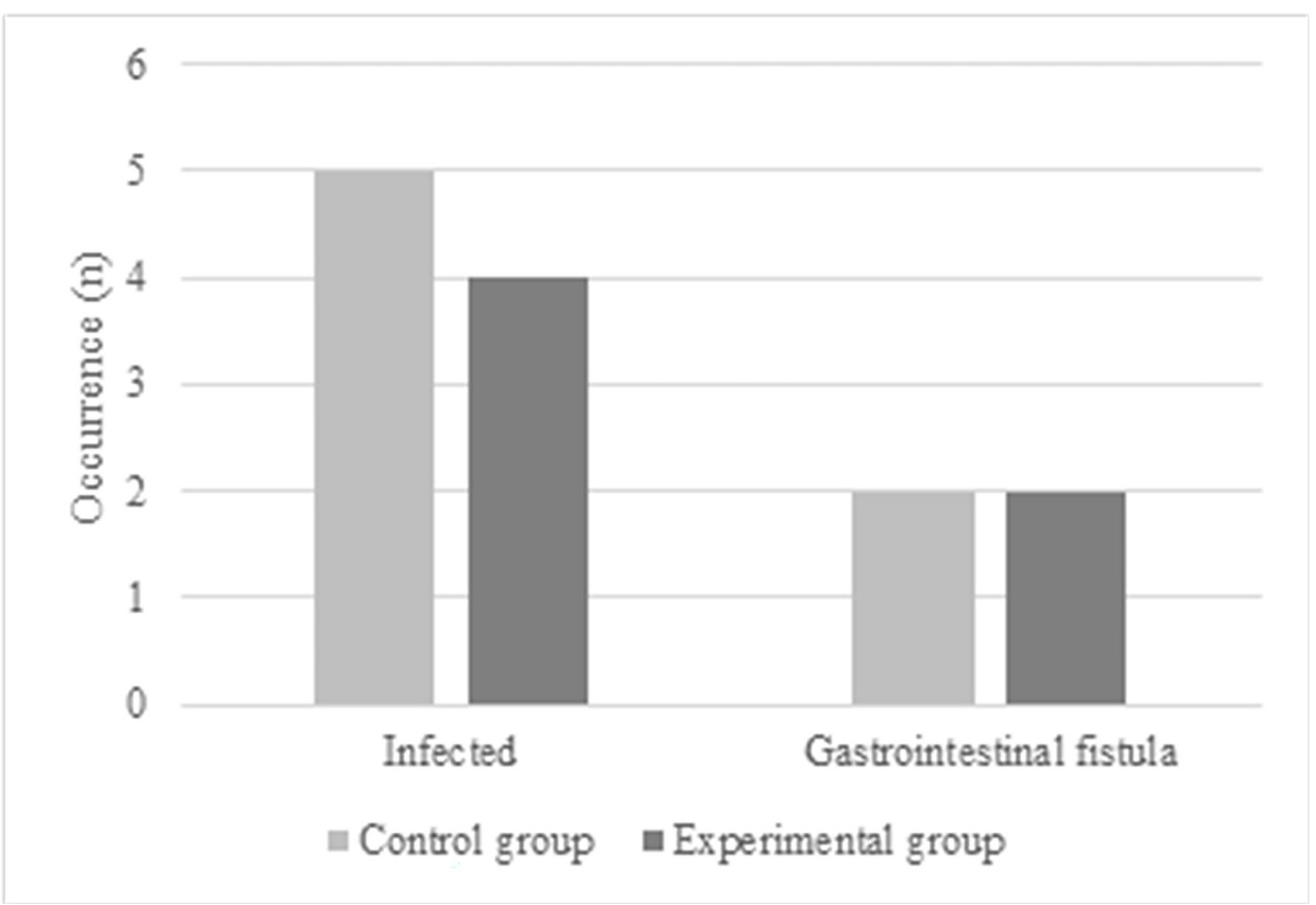

**Fig 9. Comparison of adverse events of patients with different anesthesia methods.**

Ctrl group patients ($P<0.05$). The recovery rate of postoperative pain in Exp group patients was remarkably accelerated, and postoperative TTLD was shortened ($P<0.05$). This indicates that the use of anesthesia induced dexamethasone can promote the recovery of gastrointestinal function, effectively reduce the pain of patients, accelerate the recovery of liquid diet after surgery, and promote faster recovery of patients. Chen et al. (2020) [28] showed that a single intravenous injection of 8 mg dexamethasone during anesthesia induction can significantly reduce the time of gastrointestinal recurrence, improve abdominal distension at 72 hours, and promote the tolerance of liquid diet. Wang et al. (2022) [29] also showed that the anti-inflammatory effect of dexamethasone could effectively reduce the postoperative VAS score of patients. However, the total recovery time of gastrointestinal ventilation in Ctrl group patients was longer than that in Exp group patients. This disparity could stem from variations in surgical techniques, or it may arise from the potential waning effect of a single preoperative dose over time, warranting further investigation. Additionally, patients in the Exp group exhibited significantly accelerated recovery rates of serum-related indices ($P<0.05$) and experienced a reduced mean LOS compared to the Ctrl group ($P<0.05$). The mechanisms underlying dexamethasone's action are multifaceted, potentially encompassing its anti-inflammatory properties and effects on the central nervous system. Prior research has indicated that perioperative administration of dexamethasone in conjunction with furosemide is beneficial in mitigating postoperative inflammatory responses, thereby improving clinical outcomes and shortening LOS [30].

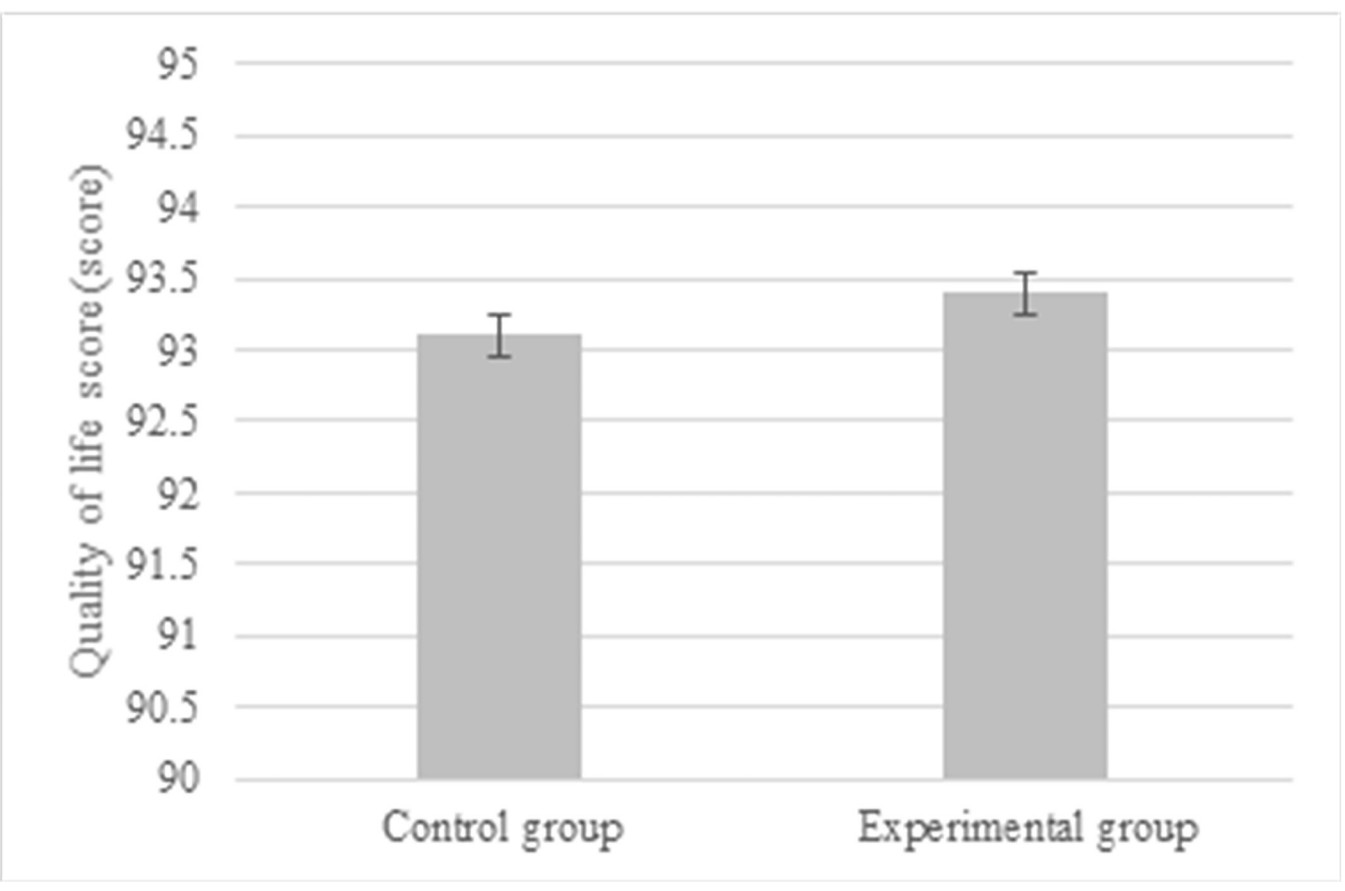

**Fig 10. Level of quality of life of patients after the surgery.**

## 5. Conclusion

This study compared the effects of deep learning-based dexamethasone pretreatment with conventional dexamethasone treatment on surgical outcomes in patients with GITs. It was found that administering dexamethasone under the anesthesia management of GITs based on GBDT and PKPD models can promote patient recovery, reduce the incidence of POCD, and improve patient prognosis. Nonetheless, the observed benefits may be influenced by the relatively small sample size, necessitating further investigation with larger cohorts. In conclusion, preoperative anesthesia pretreatment based on deep learning technology holds promise for enhancing the therapeutic outcomes of gastrointestinal cancer surgery in elderly patients, underscoring its clinical relevance and application value.

## Supporting information

**S1 Data.**
(XLSX)

## Author Contributions

**Data curation:** Kun Lu, Chun Pu.

**Formal analysis:** Kun Lu, Chun Pu, Xue Lei.

**Investigation:** Qiang Li, Chun Pu, Qiang Fu.

**Methodology:** Kun Lu, Chun Pu, Xue Lei, Qiang Fu.

**Project administration:** Kun Lu, Qiang Li, Xue Lei.

**Resources:** Kun Lu, Chun Pu, Qiang Fu.

**Software:** Kun Lu, Qiang Li, Qiang Fu.

**Supervision:** Kun Lu, Qiang Li, Chun Pu, Xue Lei.

**Validation:** Kun Lu.

**Visualization:** Kun Lu.

**Writing – original draft:** Kun Lu, Qiang Li, Chun Pu, Xue Lei, Qiang Fu.

**Writing – review & editing:** Kun Lu, Qiang Li, Chun Pu, Xue Lei, Qiang Fu.

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
