## [Editor Report · Decision Letter 0]

22 Feb 2024

PONE-D-23-40889+Effect of Dexamethasone Pretreatment Using Deep Learning on the Surgical Effect of Patients with Gastrointestinal TumorsPLOS ONE

Dear Dr. Fu,

Thank you for submitting your manuscript to PLOS ONE. After careful consideration, we feel that it has merit but does not fully meet PLOS ONE’s publication criteria as it currently stands. Therefore, we invite you to submit a revised version of the manuscript that addresses the points raised during the review process.

**Lots of English errors from Title, Abstract,...  Many incomplete sentences...****Please re-write and go through English editing!** **I should reject this manuscript upfront.  Just give the authors another chance!**

We look forward to receiving your revised manuscript.

Kind regards,

Academic Editor

PLOS ONE
---

## [Author Response · Author response to Decision Letter 0]

20 Mar 2024

PONE-D-23-40889

+Effect of Dexamethasone Pretreatment Using Deep Learning on the Surgical Effect of Patients with Gastrointestinal Tumors

PLOS ONE

Dear Dr. Fu,

Thank you for submitting your manuscript to PLOS ONE. After careful consideration, we feel that it has merit but does not fully meet PLOS ONE’s publication criteria as it currently stands. Therefore, we invite you to submit a revised version of the manuscript that addresses the points raised during the review process.

Lots of English errors from Title, Abstract,... Many incomplete sentences...

Please re-write and go through English editing!

I should reject this manuscript upfront. Just give the authors another chance!

Reply: Thank you for your suggestion. I have proof the whole article.

We look forward to receiving your revised manuscript.

Kind regards,

Academic Editor

PLOS ONE

Journal Requirements:

Reply: Thank you for your suggestion. I have adjusted the text according to the style you provided, including title format, paragraph space, chart title format, reference format, and provided TIFF format images.

Reply: Thank you for your suggestion. The above data have been provided.

Reply: Thank you for your suggestion. I have has added the name of the ethics committee in the methods section, and the signing of informed consent forms has been described in the inclusion criteria.

---

## [Decision Letter · Decision Letter 1]

2 Apr 2024

PONE-D-23-40889R1Effect of Dexamethasone Pretreatment Using Deep Learning on the Surgical Effect of Patients with Gastrointestinal TumorsPLOS ONE

Dear Dr. Fu,

Thank you for submitting your manuscript to PLOS ONE. After careful consideration, we feel that it has merit but does not fully meet PLOS ONE’s publication criteria as it currently stands. Therefore, we invite you to submit a revised version of the manuscript that addresses the points raised during the review process.

We look forward to receiving your revised manuscript.

Kind regards,

Academic Editor

PLOS ONE

**Additional Editor Comments:**

Please revise.

Reviewers' comments:

Reviewer's Responses to Questions

**Comments to the Author**

1. If the authors have adequately addressed your comments raised in a previous round of review and you feel that this manuscript is now acceptable for publication, you may indicate that here to bypass the “Comments to the Author” section, enter your conflict of interest statement in the “Confidential to Editor” section, and submit your "Accept" recommendation.

Reviewer #1: All comments have been addressed

Reviewer #2: (No Response)

2. Is the manuscript technically sound, and do the data support the conclusions?

Reviewer #1: Yes

Reviewer #2: Partly

3. Has the statistical analysis been performed appropriately and rigorously? 

Reviewer #1: Yes

Reviewer #2: No

4. Have the authors made all data underlying the findings in their manuscript fully available?

Reviewer #1: Yes

Reviewer #2: Yes

5. Is the manuscript presented in an intelligible fashion and written in standard English?

Reviewer #1: Yes

Reviewer #2: Yes

6. Review Comments to the Author

Reviewer #1: thanks

the revised paper is better than the original one

i recommends adding recent references (in the last 5 years)

your hard work is clear

for my point of view the paper now is suitable for publication.

congratulations

Reviewer #2: I have reviewed the revised submission. After going through it, I still seem to require more clarifications.

1. Please clarify two different statements written about exp and ctrl groups:

In Abstract: The participants were randomly assigned to either the control (Ctrl) group (n=40) or the experimental (Exp) group (n=40). The Ctrl group received a 2 mL intravenous injection of 0.9% saline solution 1-2 hours prior to surgical incision, while

the Exp group received a 10 mg (2 mL) intravenous injection of dexamethasone at the same time point.

While in Research Object heading: Participants were randomly assigned to two groups based on the preoperative anesthesia pretreatment methods implemented. In the control (Ctrl) group, patients received an intravenous injection of 10 mg (2 mL)

dexamethasone administered manually 1-2 hours prior to skin incision. Conversely, the experimental

(Exp) group underwent anesthesia detection utilizing deep learning technology in addition to the

anesthesia pretreatment protocol administered to the Ctrl group. these seem to be opposing statements.

2. Please clarify the use of Deep Learning algorithms as a research tool for this comparative study of two groups. You have mentioned that these were used in a sample of 10 patients. What was the rationale for not using in all patients?

3. To me as a clinician the study seems to compare the beneficial effects of pre-anesthetic use of dexamethasone on post-operative cognitive behaviour and Gastrointestinal recovery of patients. Dexamethasone affects teh neuronal networks of the body, part of our deep learning networks. Actual tests performed in both groups were the use of MoCA, GI functional tests and other lab tests. The table 1 based on Deep Learning is given for a sample of 10 patients. My query is: even if you remove the table 1and do not employ deep learning algorithms in this study, how would it affect your results? Your data is still showing the beneficial effectg of using dexamethasone. Please clarify in your introduction and methodology how was this technology used and what difference was noted in two groups.

In Conclusion: You have mentioned the clear benefits of using Deep Learning processing on post-operative recovery of patients. It seems this statement has been used as synonym for use of dexamethasone as there is no mention of the medication in concluding statements. Please clarify it, too.

In summary, please make your manuscript as elaborate, as possible, for clinicians and physicians to understand clearly.

7. PLOS authors have the option to publish the peer review history of their article (what does this mean?). If published, this will include your full peer review and any attached files.

Reviewer #1: **Yes: **hazim alhiti

Reviewer #2: No

---

## [Author Response · Author response to Decision Letter 1]

23 Apr 2024

Reviewer #1: thanks

the revised paper is better than the original one

i recommends adding recent references (in the last 5 years)

Reply: Thank you for your suggestion. I have reviewed the literature and ensured that the most recent references are added where applicable to keep our content up-to-date with the latest research findings and developments.

your hard work is clear

for my point of view the paper now is suitable for publication.

congratulations

Reviewer #2: I have reviewed the revised submission. After going through it, I still seem to require more clarifications.

1. Please clarify two different statements written about exp and ctrl groups:

In Abstract: The participants were randomly assigned to either the control (Ctrl) group (n=40) or the experimental (Exp) group (n=40). The Ctrl group received a 2 mL intravenous injection of 0.9% saline solution 1-2 hours prior to surgical incision, while

the Exp group received a 10 mg (2 mL) intravenous injection of dexamethasone at the same time point.

While in Research Object heading: Participants were randomly assigned to two groups based on the preoperative anesthesia pretreatment methods implemented. In the control (Ctrl) group, patients received an intravenous injection of 10 mg (2 mL)

dexamethasone administered manually 1-2 hours prior to skin incision. Conversely, the experimental

(Exp) group underwent anesthesia detection utilizing deep learning technology in addition to the

anesthesia pretreatment protocol administered to the Ctrl group. these seem to be opposing statements.

Reply: Thank you for providing the modification suggestions. We understand your concerns and appreciate your feedback.

The inconsistency between the description of the experimental group and the control group in the abstract and research subject title is our mistake. Here, we clarify as follows:

In our study, we did have two groups: the control group (Ctrl) and the experimental group (Exp). The description mentioned in the title of the research subjects is correct, that is, the control group received dexamethasone injection, while the experimental group received dexamethasone injection assisted by a deep learning-based anesthesia detection system.

However, the description in the abstract is inaccurate. We deeply apologize for this. The correct description should be: participants were randomly assigned to a control group and an experimental group. In the control group, patients manually received intravenous injection of dexamethasone 1-2 hours before the skin incision. On the contrary, the experimental group used deep learning techniques to monitor anesthesia status on the basis of the control group.

We thank you again for pointing out this issue and sincerely apologize for any inconvenience caused by this confusion. We have revised the title to reflect the true situation of the experiment.

2. Please clarify the use of Deep Learning algorithms as a research tool for this comparative study of two groups. You have mentioned that these were used in a sample of 10 patients. What was the rationale for not using in all patients?

Reply: Thank you for providing the modification suggestions. Regarding the application of deep learning algorithms in anesthesia detection systems evaluated using 10 patients in the study, no comparison was made between the two groups of patients. Its main purpose was to test the performance of the deep learning-based BIS detection system constructed in the study. The reason why it was not used in all patients is because deep learning algorithms usually require a large amount of computing resources and time for training and analysis. In some cases, we may not have sufficient computing resources to process data for the entire patient population. Therefore, to avoid resource waste, we choose to use these algorithms in smaller samples.

3. To me as a clinician the study seems to compare the beneficial effects of pre-anesthetic use of dexamethasone on post-operative cognitive behaviour and Gastrointestinal recovery of patients. Dexamethasone affects teh neuronal networks of the body, part of our deep learning networks. Actual tests performed in both groups were the use of MoCA, GI functional tests and other lab tests. The table 1 based on Deep Learning is given for a sample of 10 patients. My query is: even if you remove the table 1and do not employ deep learning algorithms in this study, how would it affect your results? Your data is still showing the beneficial effectg of using dexamethasone. Please clarify in your introduction and methodology how was this technology used and what difference was noted in two groups.

In Conclusion: You have mentioned the clear benefits of using Deep Learning processing on post-operative recovery of patients. It seems this statement has been used as synonym for use of dexamethasone as there is no mention of the medication in concluding statements. Please clarify it, too.

In summary, please make your manuscript as elaborate, as possible, for clinicians and physicians to understand clearly.

Reply: Thank you very much for reviewing our research and providing valuable suggestions for revisions. Here is my response to your question and corresponding modifications:

(1) Based on the above modifications, the control group was given dexamethasone, while the experimental group was given dexamethasone with the assistance of a deep learning-based anesthesia detection system. Therefore, if Table 1 is deleted and the deep learning algorithm is not used for grouping, it will not form a control analysis. In addition, in the introduction and methodology, we will provide a detailed explanation of how deep learning algorithms are applied in our research, clarify the differences between the two groups, and ensure that readers can clearly understand the technical details in the study.

(2) In the conclusion section, we will clarify that the benefits of deep learning processing are not synonymous with the use of dexamethasone. We will clarify this point and ensure that the conclusion section clearly expresses the importance of dexamethasone in postoperative recovery of patients, while not confusing the benefits of deep learning processing with the effectiveness of drug therapy.

We will write the manuscript as detailed as possible according to your suggestions to ensure that clinical doctors and doctors can clearly understand our research. Thank you again for your suggestions and review.

---

## [Decision Letter · Decision Letter 2]

13 May 2024

Effect of Dexamethasone Pretreatment Using Deep Learning on the Surgical Effect of Patients with Gastrointestinal Tumors

PONE-D-23-40889R2

Dear Dr. Fu,

We’re pleased to inform you that your manuscript has been judged scientifically suitable for publication and will be formally accepted for publication once it meets all outstanding technical requirements.

Kind regards,

Academic Editor

PLOS ONE

Additional Editor Comments (optional):

Reviewers' comments:

Reviewer's Responses to Questions

**Comments to the Author**

1. If the authors have adequately addressed your comments raised in a previous round of review and you feel that this manuscript is now acceptable for publication, you may indicate that here to bypass the “Comments to the Author” section, enter your conflict of interest statement in the “Confidential to Editor” section, and submit your "Accept" recommendation.

Reviewer #1: All comments have been addressed

Reviewer #2: All comments have been addressed

2. Is the manuscript technically sound, and do the data support the conclusions?

Reviewer #1: Yes

Reviewer #2: (No Response)

3. Has the statistical analysis been performed appropriately and rigorously? 

Reviewer #1: Yes

Reviewer #2: (No Response)

4. Have the authors made all data underlying the findings in their manuscript fully available?

Reviewer #1: Yes

Reviewer #2: (No Response)

5. Is the manuscript presented in an intelligible fashion and written in standard English?

Reviewer #1: Yes

Reviewer #2: (No Response)

6. Review Comments to the Author

Reviewer #1: bias is clear

ethical approval was clear

the topic is not unique

the paper looks sound after revision

congratulations

Reviewer #2: (No Response)

7. PLOS authors have the option to publish the peer review history of their article (what does this mean?). If published, this will include your full peer review and any attached files.

Reviewer #1: **Yes: **hazim alhiti

Reviewer #2: **Yes: **Mohammed Amir

---

## [Editor Report · Acceptance letter]

25 Jun 2024

PONE-D-23-40889R2 

PLOS ONE

Dear Dr. Fu, 

I'm pleased to inform you that your manuscript has been deemed suitable for publication in PLOS ONE. Congratulations! Your manuscript is now being handed over to our production team.

Kind regards, 

on behalf of

Dr. Robert Jeenchen Chen 

Academic Editor

PLOS ONE